# Multi-step learning and underlying structure in statistical models

**Maia Fraser**
Dept. of Mathematics and Statistics
Brain and Mind Research Institute
University of Ottawa
Ottawa, ON K1N 6N5, Canada
`mfrase8@uottawa.ca`

## Abstract

In multi-step learning, where a final learning task is accomplished via a sequence of intermediate learning tasks, the intuition is that successive steps or levels transform the initial data into representations more and more "suited" to the final learning task. A related principle arises in transfer-learning where Baxter (2000) proposed a theoretical framework to study how learning multiple tasks transforms the inductive bias of a learner. The most widespread multi-step learning approach is semi-supervised learning with two steps: unsupervised, then supervised. Several authors (Castelli-Cover, 1996; Balcan-Blum, 2005; Niyogi, 2008; Ben-David et al, 2008; Urner et al, 2011) have analyzed SSL, with Balcan-Blum (2005) proposing a version of the PAC learning framework augmented by a "compatibility function" to link concept class and unlabeled data distribution. We propose to analyze SSL and other multi-step learning approaches, much in the spirit of Baxter's framework, by defining a learning problem generatively as a joint statistical model on $X \times Y$. This determines in a natural way the class of conditional distributions that are possible with each marginal, and amounts to an abstract form of compatibility function. It also allows to analyze both discrete and non-discrete settings. As tool for our analysis, we define a notion of $\gamma$-uniform shattering for statistical models. We use this to give conditions on the marginal and conditional models which imply an advantage for multi-step learning approaches. In particular, we recover a more general version of a result of Poggio et al (2012): under mild hypotheses a multi-step approach which learns features invariant under successive factors of a finite group of invariances has sample complexity requirements that are additive rather than multiplicative in the size of the subgroups.

## 1  Introduction

The classical PAC learning framework of Valiant (1984) considers a learning problem with unknown true distribution $p$ on $X \times Y$, $Y = \{0, 1\}$ and fixed concept class $\mathcal{C}$ consisting of (deterministic) functions $f : X \to Y$. The aim of learning is to select a hypothesis $h : X \to Y$, say from $\mathcal{C}$ itself (realizable case), that best recovers $f$. More formally, the class $\mathcal{C}$ is said to be *PAC learnable* if there is a learning algorithm that with high probability selects $h \in \mathcal{C}$ having arbitrarily low generalization error for all possible distributions $D$ on $X$. The distribution $D$ governs both the sampling of points $z = (x, y) \in X \times Y$ by which the algorithm obtains a training sample and also the cumulation of error over all $x \in X$ which gives the generalization error. A modification of this model, together with the notion of *learnable with a model of probability (resp. decision rule)* (Haussler, 1989; Kearns and Schapire, 1994), allows to treat non-deterministic functions $f : X \to Y$ and the case $Y = [0, 1]$ analogously. Polynomial dependence of the algorithms on sample size and reciprocals

of probability bounds is further required in both frameworks for *efficient* learning. Not only do these frameworks consider worst case error, in the sense of requiring the generalization error to be small for arbitrary distributions $D$ on $X$, they assume the same concept class $\mathcal{C}$ regardless of the true underlying distribution $D$. In addition, *choice* of the hypothesis class is taken as part of the *inductive bias* of the algorithm and not addressed.

Various, by now classic, measures of complexity of a hypothesis space (e.g., VC dimension or Rademacher complexity, see Mohri et al. (2012) for an overview) allow to prove upper bounds on generalization error in the above setting, and distribution-specific variants of these such as annealed VC-entropy (see Devroye et al. (1996)) or Rademacher averages (beginning with Koltchinskii (2001)) can be used to obtain more refined upper bounds.

The widespread strategy of semi-supervised learning (SSL) is known not to fit well into PAC-style frameworks (Valiant, 1984; Haussler, 1989; Kearns and Schapire, 1994). SSL algorithms perform a first step using unlabeled training data drawn from a distribution on $X$, followed by a second step using labeled training data from a joint distribution on $X \times Y$. This has been studied by several authors (Balcan and Blum, 2005; Ben-David et al., 2008; Urner et al., 2011; Niyogi, 2013) following the seminal work of Castelli and Cover (1996) comparing the value of unlabeled and labeled data. One immediate observation is that without some tie between the possible marginals $D$ on $X$ and the concept class $\mathcal{C}$ which records possible conditionals $p(y|x)$, there is no benefit to unlabeled data: if $D$ can be arbitrary then it conveys no information about the true joint distribution that generated labeled data. Within PAC-style frameworks, however, $\mathcal{C}$ and $D$ are completely independent. Balcan and Blum therefore proposed augmenting the PAC learning framework by the addition of a *compatibility function* $\chi : \mathcal{C} \times \mathcal{D} \to [0, 1]$, which records the amount of compatibility we *believe* each concept from $\mathcal{C}$ to have with each $D \in \mathcal{D}$, the class of "all" distributions on $X$. This function is required to be learnable from $D$ and is then used to reduce the concept class from $\mathcal{C}$ to a sub-class which will be used for the subsequent (supervised) learning step. If $\chi$ is a good compatible function this sub-class should have lesser complexity than $\mathcal{C}$ (Balcan and Blum, 2005). While PAC-style frameworks in essence allow the true joint distribution to be anything in $\mathcal{C} \times \mathcal{D}$, the existence of a good compatibility function in the sense of Balcan and Blum (2005) implicitly assumes the joint model *that we believe in* is smaller. We return to this point in Section 2.1.

In this paper we study properties of multi-step learning strategies – those which involve multiple training steps – by considering the advantages of breaking a single learning problem into a sequence of two learning problems. We start by assuming a true distribution which comes from a class of joint distributions, i.e. statistical model, $\mathcal{P}$ on $X \times Y$. We prove that underlying structure of a certain kind in $\mathcal{P}$, together with differential availability of labeled vs. unlabeled data, imply a quantifiable advantage to multi-step learning at finite sample size. The structure we need is the existence of a representation $t(x)$ of $x \in X$ which is a sufficient statistic for the classification or regression of interest. Two common settings where this assumption holds are: manifold learning and group-invariant feature learning. In these settings we have respectively

1. $t = t_{p_X}$ is determined by the marginal $p_X$ and $p_X$ is concentrated on a submanifold of $X$,
2. $t = t_G$ is determined by a group action on $X$ and $p(y|x)$ is invariant[1] under this action.

Learning $t$ in these cases corresponds respectively to learning manifold features or group-invariant features; various approaches exist (see (Niyogi, 2013; Poggio et al., 2012) for more discussion) and we do not assume any fixed method. Our framework is also not restricted to these two settings. As a tool for analysis we define a variant of VC dimension for statistical models which we use to prove a useful lower bound on generalization error even[2] under the assumption that the true distribution comes from $\mathcal{P}$. This allows us to establish a gap *at finite sample size* between the error achievable by a single-step purely supervised learner and that achievable by a semi-supervised learner. We do not claim an asymptotic gap. The purpose of our analysis is rather to show that differential *finite* availability of data can dictate a multi-step learning approach. Our applications are respectively a strengthening of a manifold learning example analyzed by Niyogi (2013) and a group-invariant features example related to a result of Poggio et al. (2012). We also discuss the relevance of these to biological learning.

Our framework has commonalities with a framework of Baxter (2000) for transfer learning. In that work, Baxter considered learning the inductive bias (i.e., the hypothesis space) for an algorithm for a

"target" learning task, based on experience from previous "source" learning tasks. For this purpose he defined a *learning environment* $\mathcal{E}$ to be a class of probability distributions on $X \times Y$ together with an unknown probability distribution $Q$ on $\mathcal{E}$, and assumed $\mathcal{E}$ to restrict the possible joint distributions which may arise. We also make a generative assumption, assuming joint distributions come from $\mathcal{P}$, but we do not use a prior $Q$. Within his framework Baxter studied the reduction in generalization error for an algorithm to learn a new task, defined by $p \in \mathcal{E}$, when given access to a sample from $p$ and a sample from each of $m$ other learning tasks, $p_1, \ldots, p_m \in \mathcal{E}$, chosen randomly according to $Q$, compared with an algorithm having access to only a sample from $p$. The analysis produced upper bounds on generalization error in terms of covering numbers and a lower bound was also obtained in terms of VC dimension in the specific case of shallow neural networks. In proving our lower bound in terms of a variant of VC dimension we use a minimax analysis.

## 2 Setup

We assume a learning problem is specified by a joint probability distribution $p$ on $Z = X \times Y$ and a particular (regression, classification or decision) function $f_p : X \to \mathbb{R}$ determined entirely by $p(y|x)$. Moreover, we postulate a statistical model $\mathcal{P}$ on $X \times Y$ and assume $p \in \mathcal{P}$. Despite the simplified notation, $f_p(x)$ depends on the conditionals $p(y|x)$ and not the entire joint distribution $p$.

There are three main types of learning problem our framework addresses (reflected in three types of $f_p$). When $y$ is noise-free, i.e. $p(y|x)$ is concentrated at a single $y$-value $v_p(x) \in \{0, 1\}$, $f_p = v_p : X \to \{0, 1\}$ (classification); here $f_p(x) = E_p(y|x)$. When $y$ is noisy, then either $f_p : X \to \{0, 1\}$ (classification/decision) or $f_p : X \to [0, 1]$ (regression) and $f_p(x) = E_p(y|x)$. In all three cases the parameters which define $f_p$, the learning goal, depend only on $p(y|x) = E_p(y|x)$.

We assume the learner knows the model $\mathcal{P}$ and the type of learning problem, i.e., the hypothesis class is the "concept class" $\mathcal{C} := \{f_p : p \in \mathcal{P}\}$. To be more precise, for the first type of $f_p$ listed above, this is the *concept class* (Kearns and Vazirani, 1994); for the second type, it is a class of *decision rules* and for the third type, it is a class of *p-concepts* (Kearns and Schapire, 1994). For specific choice of loss functions, we seek worst-case bounds on learning rates, over all distributions $p \in \mathcal{P}$.

Our results for all three types of learning problem are stated in Theorem 3. To keep the presentation simple, we give a detailed proof for the first two types, i.e., assuming **labels are binary**. This shows how classic PAC-style arguments for discrete $X$ can be adapted to our framework where $X$ may be smooth. Extending these arguments to handle non-binary $Y$ proceeds by the same modifications as for discrete $X$ (c.f. Kearns and Schapire (1994)). We remark that in the presence of noise, better bounds can be obtained (see Theorem 3 for details) if a more technical version of Definition 1 is used but we leave this for a subsequent paper.

We define the following probabilistic version of fat shattering dimension:

**Definition 1.** *Given $\mathcal{P}$, a class of probability distributions on $X \times \{0, 1\}$, let $\gamma \in (0, 1)$, $\alpha \in (0, 1/2)$ and $n \in \mathbb{N} = \{0, 1, \ldots, \ldots\}$. Suppose there exist (disjoint) sets $S_i \subset X$, $i \in \{1, \ldots, n\}$ with $S = \cup_i S_i$, a reference probability measure $q$ on $X$, and a sub-class $\mathcal{P}_n \subset \mathcal{P}$ of cardinality $|\mathcal{P}_n| = 2^n$ with the following properties:*

1. *$q(S_i) \geq \gamma/n$ for every $i \in \{1, \ldots, n\}$*

2. *$q$ lower bounds the marginals of all $p \in \mathcal{P}_n$ on $S$, i.e. $\int_B dp_X \geq \int_B dq$ for any $p$-measurable subset $B \subset S$*

3. *$\forall\, e \in \{0, 1\}^n$, $\exists\, p \in \mathcal{P}_n$ such that $E_p(y|x) > 1/2 + \alpha$ for $x \in S_i$ when $e_i = 1$ and $E_p(y|x) < 1/2 - \alpha$ for $x \in S_i$ when $e_i = 0$*

*then we say $\mathcal{P}$ $\alpha$-**shatters** $S_1, \ldots, S_n$ $\gamma$-**uniformly using** $\mathcal{P}_n$. The $\gamma$-**uniform** $\alpha$-**shattering dimension** of $\mathcal{P}$ is the largest $n$ such that $\mathcal{P}$ $\alpha$-shatters some collection of $n$ subsets of $X$ $\gamma$-uniformly.*

This provides a measure of complexity of the class $\mathcal{P}$ of *distributions* in the sense that it indicates the variability of the expected $y$-values for $x$ constrained to lie in the region $S$ with measure at least $\gamma$ under corresponding marginals. The reference measure $q$ serves as a lower bound on the marginals and ensures that they "uniformly" assign probabilty at least $\gamma$ to $S$. Richness (variability) of conditionals is thus traded off against uniformity of the corresponding marginal distributions.

**Remark 2** (Uniformity of measure). *The technical requirement of a reference distribution $q$ is automatically satisfied if all marginals $p_X$ for $p \in \mathcal{P}_n$ are uniform over $S$. For simplicity this is the situation considered in all our examples. The weaker condition (in terms of $q$) that we postulate in Definition 1 is however sufficient for our main result, Theorem 3.*

If $f_p$ is binary and $y$ is noise-free then $\mathcal{P}$ shatters $S_1, \ldots, S_n$ $\gamma$-uniformly if and only if there is a sub-class $\mathcal{P}_n \subset \mathcal{P}$ with the specified uniformity of measure, such that each $f_p(\cdot) = E_p(y|\cdot)$, $p \in \mathcal{P}_n$ is constant on each $S_i$ and the induced set-functions shatter $\{S_1, \ldots, S_n\}$ in the usual (Vapnik-Chervonenkis) sense. In that case, $\alpha$ may be chosen arbitrarily in $(0, 1/2)$ and we omit mention of it. If $f_p$ takes values in $[0, 1]$ or $f_p$ is binary and $y$ noisy then $\gamma$-uniform shattering can be expressed in terms of fat-shattering (both at scale $\alpha$).

We show that the $\gamma$-uniform $\alpha$-shattering dimension of $\mathcal{P}$ can be used to lower bound the sample size required by even the most powerful learner of this class of problems. The proof is in the same spirit as purely combinatorial proofs of lower bounds using VC-dimension. Essentially the added condition on $\mathcal{P}$ in terms of $\gamma$ allows to convert the risk calculation to a combinatorial problem. As a counterpoint to the lower bound result, we consider an alternative two step learning strategy which makes use of underlying structure in $X$ implied by the model $\mathcal{P}$ and we obtain upper bounds for the corresponding risk.

## 2.1 Underlying structure

We assume a representation $t : X \to \mathbb{R}^k$ of the data, such that $p(y|x)$ can be expressed in terms of $p(y|t(x))$, say $f_p(x) = g_\theta(t(x))$ for some parameter $\theta \in \Theta$. Such a $t$ is generally known in Statistics as a *sufficient dimension reduction* for $f_p$ but here we make no assumption on the dimension $k$ (compared with the dimension of $X$). This is in keeping with the paradigm of feature extraction for use in kernel machines, where the dimension of $t(X)$ may even be higher than the original dimension of $X$. As in that setting, what will be important is rather that the intermediate representation $t(x)$ *reduce the complexity* of the concept space. While $t$ depends on $p$ we will assume it does so *only via $X$*. For example $t$ could depend on $p$ through the marginal $p_X$ on $X$ or possible group action on $X$; it is a manifestation in the data $X$, possibly over time, of underlying structure in the true joint distribution $p \in \mathcal{P}$. The representation $t$ captures structure in $X$ induced by $p$. On the other hand, the regression function itself depends only on the conditional $p(y|t(x))$.

In general, the natural factorization $\pi : \mathcal{P} \to \mathcal{P}_X$, $p \mapsto p_X$ determines for each marginal $q \in \mathcal{P}_X$ a collection $\pi^{-1}(q)$ of possible conditionals, namely those $p(y|x)$ arising from joint $p \in \mathcal{P}$ that have marginal $p_X = q$. More generally any sufficient statistic $t$ induces a similar factorization (c.f. Fisher-Neyman characterization) $\pi_t : \mathcal{P} \to \mathcal{P}_t$, $p \mapsto p_t$, where $\mathcal{P}_t$ is the marginal model with respect to $t$, and only conditionals $p(y|t)$ are needed for learning. As before, given a known marginal $q \in \mathcal{P}_t$, this implies a collection $\pi_t^{-1}(q)$ of possible conditionals $p(y|t)$ relevant to learning.

Knowing $q$ thus reduces the original problem where $p(y|x)$ or $p(y|t)$ can come from any $p \in \mathcal{P}$ to one where it comes from $p$ in a reduced class $\pi^{-1}(q)$ or $\pi_t^{-1}(q) \subsetneq \mathcal{P}$. Note the similarity with the assumption of Balcan and Blum (2005) that a good compatibility function *reduce* the concept class. In our case the concept class $\mathcal{C}$ consists of $f_p$ defined by $p(y|t)$ in $\cup_t \mathcal{P}_{Y|t}$ with $\mathcal{P}_{Y|t} := \{p(y|t) : p \in \mathcal{P}\}$, and marginals come from $\mathcal{P}_t$. The joint model $\mathcal{P}$ that we postulate, meanwhile, corresponds to a *subset* of $\mathcal{C} \times \mathcal{P}_t$ (pairs $(f_p, q)$ where $f_p$ uses $p \in \pi_t^{-1}(q)$). The indicator function $\chi$ for this subset is an abstract (binary) version of compatibility function (recall the compatibility function of Balcan-Blum should be a $[0, 1]$-valued function on $\mathcal{C} \times \mathcal{D}$, satisfying further practical conditions that our function typically would not). Thus, in a sense, our assumption of a joint model $\mathcal{P}$ and sufficient statistic $t$ amounts to a general form of compatibility function that links $\mathcal{C}$ and $\mathcal{D}$ without making assumptions on how $t$ might be learned. This is enough to imply the original learning problem *can* be factored into first learning the structure $t$ and then learning the parameter $\theta$ for $f_p(x) = g_\theta(t(x))$ in a reduced hypothesis space. Our goal is to understand when and why one should do so.

## 2.2 Learning rates

We wish to quantify the benefits achieved by using such a factorization in terms of the bounds on the *expected loss* (i.e. *risk*) for a sample of size $m \in \mathbb{N}$ drawn *iid* from any $p \in \mathcal{P}$. We assume the learner is provided with a sample $\bar{z} = (z_1, z_2 \cdots z_m)$, with $z_i = (x_i, y_i) \in X \times Y = Z$, drawn *iid* from the distribution $p$ and uses an algorithm $A : Z^m \to \mathcal{C} = \mathcal{H}$ to select $A(\bar{z})$ to approximate $f_p$.

Let $\ell(A(\bar{z}), f_p)$ denote a specific loss. It might be 0/1, absolute, squared, hinge or logistic loss. We define $\mathcal{L}(A(\bar{z}), f_p)$ to be the global expectation or $L^2$-norm of one of those pointwise losses $\ell$:

$$\mathcal{L}(A(\bar{z}), f_p) := E_x \ell(A(\bar{z})(x), f_p(x)) = \int_X \ell(A(\bar{z})(x), f_p(x)) dp_X(x) \tag{1}$$

or

$$\mathcal{L}(A(\bar{z}), f_p) := ||\ell(A(\bar{z}), f_p)||_{L^2(p_X)} = \sqrt{\int_X \ell(A(\bar{z})(x), f_p(x))^2 dp_X}. \tag{2}$$

Then the *worst case expected loss* (i.e. *minimax risk*) for the best learning algorithm with *no knowledge of $t_{p_X}$* is

$$R(m) := \inf_A \sup_{p \in \mathcal{P}} E_{\bar{z}} \mathcal{L}(A(\bar{z}), f_p) = \inf_A \sup_{q \in \mathcal{P}_X} \sup_{\substack{p(y|t_q) s.t. \\ p \in \mathcal{P}, p_X = q}} E_{\bar{z}} \mathcal{L}(A(\bar{z}, f_p)). \tag{3}$$

while for the best learning algorithm *with oracle knowledge of $t_{p_X}$* it is

$$Q(m) := \sup_{q \in \mathcal{P}_X} \inf_A \sup_{\substack{p(y|t_q) s.t. \\ p \in \mathcal{P}, p_X = q}} E_{\bar{z}} \mathcal{L}(A(\bar{z}, f_p)). \tag{4}$$

Some clarification is in order regarding the classes over which the suprema are taken. In principle the worst case expected loss for a given $A$ is the supremum over $\mathcal{P}$ of the expected loss. Since $f_p(x)$ is determined by $p(y|t_{p_X}(x))$, and $t_{p_X}$ is determined by $p_X$ this is a supremum over $q \in \mathcal{P}_X$ of a supremum over $p(y|t_q(\cdot))$ such that $p_X = q$. Finding the worst case expected error for the best $A$ therefore means taking the infimum of the supremum just described. In the case of $Q(m)$ since the algorithm knows $t_q$, the order of the supremum over $t$ changes with respect to the infimum: the learner can select the best algorithm $A$ using knowledge of $t_q$.

Clearly $R(m) \geq Q(m)$ by definition. In the next section, we lower bound $R(m)$ and upper bound $Q(m)$ to establish a gap between $R(m)$ and $Q(m)$.

## 3 Main Result

We show that $\gamma$-uniform shattering dimension $n$ or more implies a lower bound on the worst case expected error, $R(m)$, when the sample size $m \leq n$. In particular - in the setup specified in the previous section - if $\{g_\theta(\cdot) : \theta \in \Theta\}$ has much smaller VC dimension than $n$ this results in a distinct gap between rates for a learner with oracle access to $t_{p_X}$ and a learner without.

**Theorem 3.** *Consider the framework defined in the previous Section with $Y = \{0, 1\}$. Assume $\{g_\theta(\cdot) : \theta \in \Theta\}$ has VC dimension $d < m$ and $\mathcal{P}$ has $\gamma$-uniform $\alpha$-shattering dimension $n \geq (1+\epsilon)m$. Then, for sample size $m$, $Q(m) \leq 16\sqrt{\frac{d \log(m+1) + \log 8 + 1}{2m}}$ while $R(m) > \epsilon bc\gamma^{m+1}/8$ where $b$ depends both on the type of loss and the presence of noise, while $c$ depends on noise.*

*Assume the standard definition in (1). If $f_p$ are binary (in the noise-free or noisy setting) $b = 1$ for absolute, squared, 0-1, hinge or logistic loss. In the noisy setting, if $f_p = E(y|x) \in [0, 1]$, $b = \alpha$ for absolute loss and $b = \alpha^2$ for squared loss. In general, $c = 1$ in the noise-free setting and $c = (1/2 + \alpha)^m$ in the noisy setting. By requiring $\mathcal{P}$ to satisfy a stronger notion of $\gamma$-uniform $\alpha$-shattering one can obtain $c = 1$ even in the noisy case.*

Note that for sample size $m$ and $\gamma$-uniform $\alpha$-shattering dimension $2m$, we have $\epsilon = 1$, so the lower bound in its simplest form becomes $\gamma^{m+1}/8$. This is the bound we will use in the next Section to derive implications of Theorem 3.

**Remark 4.** *We have stated in the Theorem a simple upper bound, sticking to $Y = \{0, 1\}$ and using VC dimension, in order to focus the presentation on the lower bound which uses the new complexity measure. The upper bound could be improved. It could also be replaced with a corresponding upper bound assuming instead $Y = [0, 1]$ and fat shattering dimension $d$.*

*Proof.* The upper bound on $Q(m)$ holds for an ERM algorithm (by the classic argument, see for example Corollary 12.1 in Devroye et al. (1996)). We focus here on the lower bound for $R(m)$. Moreover, we stick to the simpler definition of $\gamma$-uniform shattering in Definition 1 and omit proof of the final statement of the Theorem, which is slightly more involved. We let $n = 2m$ (i.e. $\epsilon = 1$) and

we comment in a footnote on the result for general $\epsilon$. Let $S_1, \ldots, S_{2m}$ be sets which are $\gamma$-uniformly $\alpha$-shattered using the family $\mathcal{P}_{2m} \subset \mathcal{P}$ and denote their union by $S$. By assumption $S$ has measure at least $\gamma$ under a reference measure $q$ which is dominated by all marginals $p_X$ for $p \in \mathcal{P}_{2m}$ (see Definition 1). We divide our argument into three parts.

**1.** If we prove a lower bound for the average over $\mathcal{P}_{2m}$,

$$\forall A, \; \frac{1}{2^{2m}} \sum_{p \in \mathcal{P}_{2m}} E_{\bar{z}} \mathcal{L}(A(\bar{z}), f_p) \geq bc\gamma^{m+1}/8 \tag{5}$$

it will also be a lower bound for the supremum over $\mathcal{P}_{2m}$:

$$\forall A, \; \sup_{p \in \mathcal{P}_{2m}} E_{\bar{z}} \mathcal{L}(A(\bar{z}), f_p) \geq bc\gamma^{m+1}/8 \;.$$

and hence for the supremum over $\mathcal{P}$. It therefore suffices to prove (5).

**2.** Given $x \in S$, define $v_p(x)$ to be the more likely label for $x$ under the joint distribution $p \in \mathcal{P}_{2m}$. This notation extends to the noisy case the definition of $v_p$ already given for the noise-free case. The uniform shattering condition implies $p(v_p(x)|x) > 1/2 + \alpha$ in the noisy case and $p(v_p(x)|x) = 1$ in the noise-free case. Given $\bar{x} = (x_1, \ldots, x_m) \in S^m$, write $\bar{z}_p(\bar{x}) := (z_1, \ldots, z_m)$ where $z_j = (x_j, v_p(x_j))$. Then

$$E_{\bar{z}} \mathcal{L}(A(\bar{z}), f_p) = \int_{Z^m} \mathcal{L}(A(\bar{z}), f_p) dp^m(\bar{z})$$

$$\geq \int_{S^m \times Y^m} \mathcal{L}(A(\bar{z}), f_p) dp^m(\bar{z}) \geq c \int_{S^m} \mathcal{L}(A(\bar{z}_p(\bar{x})), f_p) dp_X^m(\bar{x})$$

where $c$ is as specified in the Theorem. Note the sets

$$V_l := \{\bar{x} \in S^m \subset X^m : \text{ the } x_j \text{ occupy exactly } l \text{ of the } S_i\}$$

for $l = 1, \ldots, m$ define a partition of $S^m$. Recall that $dp_X \geq dq$ on $S$ for all $p \in \mathcal{P}_{2m}$ so

$$\int_{S^m} \mathcal{L}(A(\bar{z}_p(\bar{x})), f_p) dp_X^m(\bar{x}) \geq \frac{1}{2^{2m}} \sum_{p \in \mathcal{P}_{2m}} \sum_{l=1}^{m} \int_{\bar{x} \in V_l} \mathcal{L}(A(\bar{z}_p(\bar{x})), f_p) \, dq^m(\bar{x})$$

$$= \sum_{l=1}^{m} \int_{\bar{x} \in V_l} \left( \underbrace{\frac{1}{2^{2m}} \sum_{p \in \mathcal{P}_{2m}} \mathcal{L}(A(\bar{z}_p(\bar{x})), f_p)}_{I} \right) dq^m(\bar{x}).$$

We claim the integrand, $I$, is bounded below by $b\gamma/8$ (this computation is performed in part 3, and depends on knowing $\bar{x} \in V_l$). At the same time, $S$ has measure at least $\gamma$ under $q$ so

$$\sum_{l=1}^{m} \int_{\bar{x} \in V_l} dq^m(\bar{x}) = \int_{\bar{x} \in S^m} dq^m(\bar{x}) \geq \gamma^m$$

which will complete the proof of (5).

**3.** We now assume a fixed but arbitrary $\bar{x} \in V_l$ and prove $I \geq b\gamma/8$. To simplify the discussion, we will refer to sets $S_i$ which contain a component $x_j$ of $\bar{x}$ as $S_i$ with data. We also need notation for the elements of $\mathcal{P}_{2m}$: for each $L \subset [2m]$ denote by $p^{(L)}$ the unique element of $\mathcal{P}_{2m}$ such that $v_{p^{(L)}}|_{S_i} = 1$ if $i \in L$, and $v_{p^{(L)}}|_{S_i} = 0$ if $i \notin L$. Now, let $L_{\bar{x}} := \{i \in [2m] : \bar{x} \cap S_i \neq \emptyset\}$. These are the indices of sets $S_i$ with data. By assumption $|L_{\bar{x}}| = l$, and so $|L_{\bar{x}}^c| = 2m - l$.

Every subset $L \subset [2m]$ and hence every $p \in \mathcal{P}_{2m}$ is determined by $L \cap L_{\bar{x}}$ and $L \cap L_{\bar{x}}^c$. We will collect together all $p^{(L)}$ having the same $L \cap L_{\bar{x}}$, namely for each $D \subset L_{\bar{x}}$ define

$$\mathcal{P}_D := \{p^{(L)} \in \mathcal{P}_{2m} : L \cap L_{\bar{x}} = D\}.$$

These $2^l$ families partition $\mathcal{P}_{2m}$ and in each $\mathcal{P}_D$ there are $2^{2m-l}$ probability distributions. Most importantly, $\bar{z}_p(\bar{x})$ is the same for all $p \in \mathcal{P}_D$ (because $D$ determines $v_p$ on the $S_i$ with data). This

implies $A(\bar{z}_p(\bar{x})) : X \to \mathbb{R}$ is the *same* function[3] of $X$ for all $p$ in a given $\mathcal{P}_D$. To simplify notation, since we will be working within a single $\mathcal{P}_D$, we write $f := A(\bar{z}(\bar{x}))$.

While $f$ is the hypothesized regression function given data $\bar{x}$, $f_p$ is the true regression function when $p$ is the underlying distribution. For each set $S_i$ let $v_i$ be 1 if $f$ is above $1/2$ on a majority of $S_i$ using reference measure $q$ (a $q$-majority) and 0 otherwise.

We now focus on the "unseen" $S_i$ where no data lie (i.e., $i \in L_{\bar{x}}^c$) and use the $v_i$ to specify a 1-1 correspondence between elements $p \in \mathcal{P}_D$ and subsets $K \subset L_{\bar{x}}^c$:

$$p \in \mathcal{P}_D \quad \longleftrightarrow \quad K_p := \{i \in L_{\bar{x}}^c : v_p \neq v_i\}.$$

Take a specific $p \in \mathcal{P}_D$ with its associated $K_p$. We have $|f(x) - f_p(x)| > \alpha$ on the $q$-majority of the set $S_i$ for all $i \in K_p$.

The condition $|f(x) - f_p(x)| > \alpha$ with $f(x)$ and $f_p(x)$ on opposite sides of $1/2$ implies a lower bound on $\ell(f(x), f_p(x))$ for each of the pointwise loss functions $\ell$ that we consider (0/1, absolute, square, hinge, logistic). The value of $b$, however, differs from case to case (see Appendix).

For now we have,

$$\int\limits_{S_i} \ell(f(x), f_p(x)) \, dp_X(x) \geq \int\limits_{S_i} \ell(f(x), f_p(x)) \, dq(x) \geq b \, \frac{1}{2} \int\limits_{S_i} dq(x) \geq \frac{b\gamma}{4m}.$$

Summing over all $i \in K_p$, and letting $k = |K_p|$, we obtain (still for the same $p$)

$$\mathcal{L}(f(x), f_p(x)) \geq k \frac{b\gamma}{4m}$$

(assuming $L$ is defined by equation (1))[4]. There are $\binom{2m-\ell}{k}$ possible $K$ with cardinality $k$, for any $k = 0, \ldots, 2m - \ell$. Therefore,

$$\sum_{p \in \mathcal{P}_D} \mathcal{L}(f(x), f_p(x)) \geq \sum_{k=0}^{2m-\ell} \binom{2m-\ell}{k} k \, \frac{b\gamma}{4m} = \frac{2^{2m-\ell}(2m-\ell)}{2} \, \frac{b\gamma}{4m} \geq 2^{2m-\ell} \frac{b\gamma}{8}$$

(using $2m - \ell \geq 2m - m = m$)[5]. Since $D$ was an arbitrary subset of $L_{\bar{x}}$, this same lower bound holds for each of the $2^\ell$ families $\mathcal{P}_D$ and so

$$I = \frac{1}{2^{2m}} \sum_{p \in \mathcal{P}_{2m}} \mathcal{L}(f(x), f_p(x)) \geq \frac{b\gamma}{8}.$$

$\square$

In the constructions of the next Section it is often the case that one can prove a different level of shattering for different $n$, namely $\gamma(n)$-uniform shattering of $n$ subsets for various $n$. The following Corollary is an immediate consequence of the Theorem for such settings. We state it for binary $f_p$ without noise.

**Corollary 5.** *Let $C \in (0,1)$ and $M \in \mathbb{N}$. If $\mathcal{P}$ $\gamma(n)$-uniformly $\alpha$-shatters $n$ subsets of $X$ and $\gamma(n)^{n+1}/8 > C$ for all $n < M$ then no learning algorithm can achieve worst case expected error below $\alpha C$, using a training sample of size less than $M/2$. If such uniform shattering holds for all $n \in \mathbb{N}$ then the same lower bound applies regardless of sample size.*

Even when $\gamma(n)$-uniform shattering holds for all $n \in \mathbb{N}$ and $\lim_{n \to \infty} \gamma(n) = 1$, if $\gamma(n)$ approaches 1 sufficiently slowly then it is possible $\gamma(n)^{n+1} \to 0$ and there is no asymptotic obstacle to learning. By contrast, the next Section shows an extreme situation where $\lim_{n \to \infty} \gamma(n)^{n+1} \geq e > 0$. In that case, learning is impossible.

## 4 Applications and conclusion

**Manifold learning** We now describe a simpler, finite dimensional version of the example in Niyogi (2013). Let $X = \mathbb{R}^D$, $D \geq 2$ and $Y = \{0, 1\}$. Fix $N \in \mathbb{N}$ and consider a very simple type of 1-dimensional manifold in $X$, namely the union of $N$ linear segments, connected in circular fashion (see Figure 1). Let $\mathcal{P}_X$ be the collection of marginal distributions, each of which is supported on and assigns uniform probability along a curve of this type. There is a 1-1 correspondence between the elements of $\mathcal{P}_X$ and curves just described.

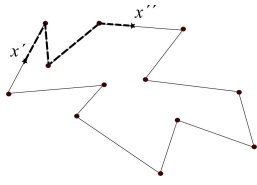

Figure 1: An example of $\mathcal{M}$ with $N = 12$. The dashed curve is labeled 1, the solid curve 0 (in next Figure as well).

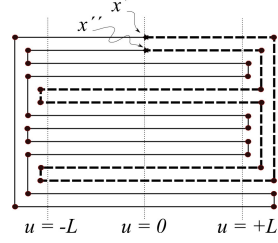

Figure 2: $\mathcal{M}$ with $N = 28 = 4(n+1)$ pieces, used to prove uniform shattering of $n$ sets (shown for the case $n = 6$ with $e = 010010$).

On each curve $\mathcal{M}$, choose two distinct points $x', x''$. Removing these disconnects $\mathcal{M}$. Let one component be labeled 0 and the other 1, then label $x'$ and $x''$ oppositely. Let $\mathcal{P}$ be the class of joint distributions on $X \times Y$ with conditionals as described and marginals in $\mathcal{P}_X$. This is a noise-free setting and $f_p$ is binary. Given $\mathcal{M}$ (or circular coordinates on $\mathcal{M}$), consider the reduced class $\mathcal{P}' := \{p \in \mathcal{P} : \text{support}(p_X) = \mathcal{M}\}$. Then $\mathcal{H}' := \{f_p : p \in \mathcal{P}'\}$ has VC dimension 3. On the other hand, for $n < N/4 - 1$ it can be shown that $\mathcal{P}$ $\gamma(n)$-uniformly shatters $n$ sets with $f_p$, where $\gamma(n) = 1 - \frac{1}{n+1}$ (see Appendix and Figure 2). Since $(1 - \frac{1}{n+1})^{n+1} \to e > 0$ as $n \to \infty$, it follows from Corollary 5 that the worst case expected error is bounded below by $e/8$ for any sample of size $n \le N/8 - 1/2$. If many linear pieces are allowed (i.e. $N$ is high) this could be an impractical number of labeled examples. By contrast with this example, $\gamma(n)$ in Niyogi's example cannot be made arbitrarily close to 1.

**Group-invariant features**   We give a simplified, partially-discrete example (for a smooth version and Figures, see Appendix). Let $Y = \{0, 1\}$ and let $X = J \times I$ where $J = \{0, 1, \ldots, n_1 - 1\} \times \{0, 1, \ldots, n_2 - 1\}$ is an $n_1$ by $n_2$ grid ($n_i \in \mathbb{N}$) and $I = [0, 1]$ is a real line segment. One should picture $X$ as a rectangular array of vertical sticks. Above each grid point $(j_1, j_2)$ consider two special points on the stick $I$, one with $i = i_+ := 1 - \epsilon$ and the other with $i = i_- := 0 + \epsilon$. Let $\mathcal{P}_X$ contain only the uniform distribution on $X$ and assume the noise-free setting. For each $\bar{e} \in \{+, -\}^{n_1 n_2}$, on each segment $(j_1, j_2) \times I$ assign, via $p_{\bar{e}}$, the label 1 above the special point (determined by $\bar{e}$) and 0 below the point. This determines a family of $n_1 n_2$ conditional distributions and thus a family $\mathcal{P} := \{p_{\bar{e}} : \bar{e} \in \{+, -\}^{n_1 n_2}\}$ of $n_1 n_2$ joint distributions. The reader can verify that $\mathcal{P}$ has $2\epsilon$-uniform shattering dimension $n_1 n_2$. Note that when the true distribution is $p_{\bar{e}}$ for some $\bar{e} \in \{+, -\}^{n_1 n_2}$ the labels will be invariant under the action $a_{\bar{e}}$ of $\mathbb{Z}_{n_1} \times \mathbb{Z}_{n_2}$ defined as follows. Given $(z_1, z_2) \in \mathbb{Z}_{n_1} \times \mathbb{Z}_{n_2}$ and $(j_1, j_2) \in J$, let the group element $(z_1, z_2)$ move the vertical stick at $(j_1, j_2)$ to the one at $(z_1 + j_1 \mod n_1, z_2 + j_2 \mod n_2)$ without flipping the stick over, just stretching it as needed so the special point $i_\pm$ determined by $\bar{e}$ on the first stick goes to the one on the second stick. The orbit space of the action can be identified with $I$. Let $t : X \times Y \to I$ be the projection of $X \times Y$ to this orbit space, then there is an induced labelling of this orbit space (because labels were invariant under the action of the group). Given access to $t$, the resulting concept class has VC dimension 1. On the other hand, given instead access to a projection $s$ for the action of the subgroup $\mathbb{Z}_{n_1} \times \{0\}$, the class $\widetilde{\mathcal{P}} := \{p(\cdot|s) : p \in \mathcal{P}\}$ has $2\epsilon$-uniform shattering dimension $n_2$. Thus we have a general setting where the over-all complexity requirements for two-step learning are $n_1 + n_2$ while for single-step learning they are $n_1 n_2$.

**Conclusion**   We used a notion of uniform shattering to demonstrate both manifold learning and invariant feature learning situations where learning becomes impossible unless the learner has access to very large amounts of labeled data or else uses a two-step semi-supervised approach in which suitable manifold- or group-invariant features are learned first in unsupervised fashion. Our examples also provide a complexity manifestation of the advantages, observed by Poggio and Mallat, of forming intermediate group-invariant features according to sub-groups of a larger transformation group.

**Acknowledgements**   The author is deeply grateful to Partha Niyogi for the chance to have been his student. This paper is directly inspired by discussions with him which were cut short much too soon. The author also thanks Ankan Saha and Misha Belkin for very helpful input on preliminary drafts.

## Footnotes

[1] This means there is a group $G$ of transformations of $X$ such that $p(y|x) = p(y|g{\cdot}x)$ for all $g \in G$.

[2] (distribution-specific lower bounds are by definition weaker than distribution-free ones)

[3]Warning: $f$ need not be an element of $\{f_p : p \in \mathcal{P}_{2n}\}$; we only know $f \in \mathcal{H} = \{f_p : p \in \mathcal{P}\}$.

[4]In the $L^2$ version, using $\sqrt{x} \geq x$, the reader can verify the same lower bound holds.

[5]In the case where we use $(1 + \epsilon)m$ instead of $2m$, we would have $(1 + \epsilon)m - \ell \geq \epsilon m$ here.

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
