[Supplementary Material]

# Appendix

Figure 3: $S^1$-orbits which are graphs of smooth functions from $J$ to $I$ (arrows indicate sides which are identified since $J = S^1$).

Figure 4: The shaded region is $B$, the union of thin strips. The thick dotted curve shows an example of $\varphi_L$ for $L = \{2\} \subset [2m]$ with $m = 2$; the resulting labels 0 and 1 on the $S_i \subset B^c$ are indicated.

## A  Comments regarding losses

When "regression" functions are binary, the 0-1, absolute and squared losses are the same $\ell_{\text{abs}}(f(x), f_p(x)) = |f(x) - f_p(x)| = |f(x) - f_p(x)|^2 = \ell_{\text{sq}}(f(x), f_p(x))$. In fact in the proof of Theorem 1 we only care about $x$ in the $q$-majority of $S_i$ for $i \in K_p$. For such $x$ the above expression has value 1. To study the hinge loss and logistic loss, assume $x$ as before. Because $f_p$ and $f$ have values on the opposite side of $1/2$ for such $x$, if we convert binary labels to labels in $\{-1, +1\}$ via the transformation $y \mapsto 2y - 1$, and convert regression functions accordingly, then the converted $f(x)$ and $f_p(x)$ will have opposite sign. This means $s := f(x)f_p(x) \leq 0$ so the hinge loss $\ell_{\text{hinge}}(f(x), f_p(x)) = 1 - s = |1 - s|$ for these $x$. This is a decreasing function of $s$ with value 1 for $s = 0$, so $\ell_{\text{hinge}}(f(x), f_p(x)) \geq 1$ for these $x$ and similarly for the logistic loss, $\ell_{\text{log}}(f(x), f_p(x)) \geq 1$. Thus $b = 1$ in all the above cases. When $f_p$ is not binary, the absolute and square losses have the same form as given above but we only know they are bounded below by $b = \alpha$ and $b = \alpha^2$ respectively.

## B  Smooth group-invariant feature example in detail

Let $X = J \times I$ where $I = [0, 1] \subset \mathbb{R}$ and $J = S^1 = \mathbb{R}/\mathbb{Z}$ (i.e. $[0, 1]$ "with the endpoints identified"). Let $Y = \{0, 1\}$. We are interested in joint probability distributions $p$ on $X \times Y$ for which $p(y|x)$ is invariant under an $S^1$-action on $X$ and for which the orbit space $X/S^1$ can be parametrized by a real variable $t(x)$. In this case $p(y|x)$ is completely determined by $t(x)$ and therefore the regression function $f_p(x) := E_p(y|x)$ is as well.

In order to obtain a simple expression for $t(x)$ we restrict to $S^1$-actions on $X$ for which orbits are graphs of smooth functions from $J$ to $I$ foliating $J \times I$. See Figure 3. In this case the projection map $t : X \to I$ which sends each $x \in X$ to the $I$-value $t(x)$ where its orbit crosses the $I$-axis gives a convenient parametrization of $X/S^1$. It is determined by the $S^1$-action.

Fix $n \in \mathbb{N}$. For each integer $k = 0, \ldots, 2n$ consider a thin strip containing the line segment $\{\frac{k}{n} - 1\} \times I$. Take two more thin strips containing respectively the curves $J \times \{0\}$ and $J \times \{1\}$. Denote by $B$ the union of all these strips (see Figure 4) and let $\beta = \int_B p_X$. By making the strips thin enough we may arrange that $\beta$ be arbitrarily small. By keeping their thickness constant (on each side of the core segment) we arrange that $\int_{B \cap (J_k \times I)} p_X = \beta/2n$ for every $k = 0, \ldots, 2n$, where $J_k$ denotes the interval $[\frac{k-1}{n} - 1, \frac{k}{n} - 1] \subset J$.

For each subset $L \subset \{1, \ldots, 2n\}$ define the step function $\phi_L : J \to I$ to be the indicator function for $\bigcup_{k \in L} J_k$. Note this is a function from $J$ to $I$ and its graph is a union of line segments parallel to the $J$-axis. Choose $\varphi_L$ to be a smooth function approximating $\phi_L$ whose graph agrees with that of $\phi_L$ outside of $B$ and such that $\varphi_L(0) = \varphi_L(1) = .5$ and $\varphi_L$ never takes on the values 1 and 0. In other words, to obtain $\varphi_L$ we smooth discontinuities of the graph of $\phi_L$ (within $B$), drag its $I$-intercept to

.5 and push it slightly away from the edges $J \times \{0\}$ and $J \times \{1\}$. Now, for any $\theta \in (0, 1)$, construct an $S^1$-action $\cdot_{L,\theta}$ on $X = J \times I$ for which the orbit through $(0, \theta)$ is the graph of $\varphi_L$ and the orbits of points $(0, i)$ are graphs of smooth functions. How these functions are chosen is not so important; one can use any foliation of the complement of the graph of $\varphi_L$ with curves whose tangent vectors are bounded away from the $I$-direction. Let the $S^1$-action along these curves have unit velocity in the $J$-direction (i.e. the action of $t \in S^1$ will send $(j, i)$ to $(j + t, i')$ where $i'$ can be read off by following the orbit through $(j, i)$ until it crosses the vertical line at $j + t$). The projection function for this action will be denoted $t_{L,\theta}$.

Denote by $p^{(L,\theta)}$ the probability distribution on $X \times Y$ having a uniform marginal $p_X^{(L,\theta)}$ on $X$ and having conditionals $p^{(L,\theta)}(y|x)$ each concentrated at a point in $Y$ as follows:

$$p^{(L,\theta)}(y|x) = \begin{cases} 1 & \text{if } t_{L,\theta}(x) < \theta \text{ and } y = 0 \\ & \text{or } t_{L,\theta}(x) \geq \theta \text{ and } y = 1 \\ 0 & \text{otherwise.} \end{cases} \tag{6}$$

Write
$$\mathcal{P} := \{p^{(L,\theta)} : L \subset \{1, \ldots, 2n\}, \theta \in (0, 1)\}.$$

For each $p \in \mathcal{P}$, $f_p(x) := E_p(y|x)$ is constant on any orbit of the $S^1$ action used to define $p$. It therefore induces on $X/S^1$ a function we denote $g_\theta(t)$. Observe that $\{g_\theta : \theta \in (0, 1)$ has VC-dimension 2 so this regression problem can be solved quickly by an ERM-type algorithm. Having oracle access to $S^1$-invariant features (for the underlying $S^1$-action) would allow labeled data $(x_i, y_i)$ for the original regression problem to be converted to data for this simpler problem. Theorem 3 allows one to quantify *how much harder* the original problem is, in terms of its $\gamma$-uniform shattering dimension. We claim that $\mathcal{P}$ has $(1 - \beta)$-uniform shattering dimension at least $2n$.

Let $S_k$ be the $k$'th strip outside of $B$:
$$S_k := (J_k \setminus B) \times I.$$

This is the $k$'th unshaded strip in Figure 4. For all $L \subset \{1, \ldots, 2n\}$, $\theta \in (0, 1)$, and $k \in \{1, \ldots, 2n\}$, $p^{(L,\theta)}(S_k) > (1 - \beta)/2n$. Moreover, $p^{(L,.5)}$ assigns to all $x \in S_k$ the value 1 if $k \in L$ and the value 0 if $k \notin L$. Defining
$$\mathcal{P}_n := \{p^{(L,.5)} \mid L \subset \{1, \ldots, 2n\}\} \subset \mathcal{P}$$

we achieve the desired $(1 - \beta)$-uniform shattering.

## C   Further discussion

We remark that our paper has avoided the term hierarchical learning. While successive representations of input data constitute a hierarchy, in practice hierarchical models are often trained using a single set of training data. To emphasize that we consider instead a *succession of learning tasks*, i.e. multiple training steps each with their respective data, we use the terminology multi-step. Also, while our focus is in characterizing learning problems that are best addressed with multi-step algorithms in terms of underlying structure in a statistical model and differential data availability, this understanding in turn complements results in neuroscience concerning the hierarchical order in which "learning" occurs in the brain. On the one hand, visual plasticity (after the developmental stage) seems to occur in top-down fashion, with changes in higher regions of the visual system occurring first and then lower ones only as needed (Ahissar and Hochstein, 2004). On the other hand, during development, primary sensory regions of the cortical system seem to develop earlier than higher ones (Bourne and Rosa, 2006). It is interesting to consider this from the point of view of our results: before any representations at lower level have been learned there may be no possibility to learn higher level concepts and gradual bottom-up development would ensure lower levels are trained first and higher levels receive input that has been transformed via the acquired representations. The long-term statistical model governing input of a specific sensory kind (say visual) will, however, in many ways not change much with time. By contrast, higher levels will receive input that has been held in memory for varying amounts of time and may also arise from different types of sensory input (or even be subject to will). The complexities of recurrence and the combinatorics inherent in combining many different source distributions could therefore be expected to produce changing learning environments at a higher level as time goes on, with corresponding new requirements for incoming representations to these high levels - a possible explanation for top-down plasticity.

The work of Niyogi (2013) focused on semi-supervised use of manifold regularization (Belkin et al., 2006). Many other examples of semi-supervised learning have been introduced and thoroughly studied. In addition the role of layers or steps in developing good representations has been explored empirically in several works on Deep Belief Networks (DBN's) and theoretical support for the manifold learning properties of autoencoders can be found in Alain and Bengio (2012). There is thus wide interest in understanding what governs the need for intermediate representations. In our manifold learning example we provide an extreme case where single-step learning is impossible. Our result on group invariant features considers a simple example where a product group acts on data and we show the sample complexity required to learn in a single step is multiplicative in the sizes of the groups, whereas for a two-step approach involving invariant features it is additive. This result is related to two separate lines of work which motivate sequences of representations via successive factors of transformation groups (Poggio et al. (2012); Mallat (2011), respectively in the human nervous system and in artificial networks). In addition, in the presence of "natural" group actions, when data are produced in time sequence, samples tend to lie along orbits of the action. This provides a powerful semi-supervised method for learning invariant features which has been increasingly exploited. Work making use of temporal data in DBN's also exists; see Poggio et al. (2012) for a thorough treatment of the idea and links to relevant papers.