[Reviews · NeurIPS 2016]

Reviewer 1

Summary

This paper provides a theoretical framework for understanding how unlabeled data can improve the performance of supervised learning. It is assumed that the function to be learned ( the regression function in the case of regression, or the posterior class probability function in the case of classification) is completely determined by the marginal distribution of X. furthermore, it is assumed that the function that maps marginal distributions on X to the desired decision function is *known*. A theorem is established that quantifies the improvement in performance compared to the case where this mapping is not known. The theory introduces a novel notion of complexity, the so-called uniform shattering dimension. The theory is illustrated very briefly a couple of more concrete scenarios.

Qualitative Assessment

Overall this is a good paper, and I would have no problem with it being accepted into the program. The theoretical results are substantial technically speaking. The work provides additional theoretical understanding to the problem of semi-supervised learning. The primary weakness of the paper is the lack of clarity in some of the presentation. Here are some examples of what I mean. 1) l 63, refers to a "joint distribution on D x C". But C is a collection of classifiers, so this framework where the decision functions are random is unfamiliar. 2) In the first three paragraphs of section 2, the setting needs to be spelled out more clearly. It seems like the authors want to receive credit for doing something in greater generality than what they actually present, and this muddles the exposition. 3) l 123, this is not the definition of "dominated" 4) for the third point of definition one, is there some connection to properties of universal kernels? See in particular chapter 4 of Steinwart and Christmann which discusses the ability of universal kernels two separate an arbitrary finite data set with margin arbitrarily close to one. 5) an example and perhaps a figure would be quite helpful in explaining the definition of uniform shattering. 6) in section 2.1 the phrase "group action" is used repeatedly, but it is not clear what this means. 7) in the same section, the notation {\cal P} with a subscript is used several times without being defined. 8) l 196-7: this requires more explanation. Why exactly are the two quantities different, and why does this capture the difference in learning settings? ---- I still lean toward acceptance. I think NIPS should have room for a few "pure theory" papers.

Confidence in this Review

2-Confident (read it all; understood it all reasonably well)


Reviewer 2

Summary

The paper studies prediction problems (a binary realizable, separable classification) within a standard PAC framework and tries to quantifies benefits which a learning algorithm can achieve by using a certain structure of the unknown data-generating distribution $P(X,Y)$. More precisely, the authors consider problems where (a) the conditional probability $P(Y=1|X=x)$ function depends on the input $x$ only through a sufficient statistic (or the *representation*) $t(x)$ and (b) function $t(.)$ depends on the unknown distribution $P(X,Y)$ only through its marginal $P(X)$. The authors approach this problem by providing (1) the upper bound on the generalization error of the "oracle learner" having access to the structure and (2) the lower bound for the performance of a standard learner with no knowledge regarding the structure in $P(X,Y)$ (Theorem 3). The lower bound is based on a novel extension of a fat-shattering dimension (Definition 1), which measures the complexity of a set of *distributions*, while the upper bound trivially follows from the standard worst-case bounds on the ERM algorithm. Finally, the authors consider two different particular examples where the oracle learner can effectively solve the prediction problem, while the standard learner fails to do so (Section 4). I checked all the proofs appearing in the main part of the paper and they are correct.

Qualitative Assessment

While the main topic of the paper seems to be important and interesting, I was not impressed with the presented results. First of all, the assumption on the structure of $P$ is too strong for my taste: not only did we assume that such a factorization holds, but we also assumed that the oracle learner knows *exactly* the sufficient statistic $t$. I think this is a rather unrealistic situation. Second, the main theoretical result (Theorem 3) trivially follows the standard proofs of the lower bounds in the supervised learning (Chapter 14 in Devroye et al.). The only difference is captured in the extension of the fat-shattering dimension of Definition 1, but apart from that these are all well-known steps: let a marginal be concentrated on shattered points, then use probabilistic method, and play with the labels on the points not appearing in the training sample. In other words, I would say Theorem 3 does not contain any significant novel ideas. Finally, Section 4 looks confusing to me: the authors did not provide the original example of Niyogi in the "Manifold Learning" paragraph, which makes it hard to assess the novelty of the current paper in this context; "Group-Invariant features" paragraph is overly dense and is not supported with clear and intuitive discussions. Concluding, I am not sure if this work provides a sufficient contribution to the literature (neither in terms of the techniques used, nor in terms of the results proved and conclusions made). DETAILED COMMENTS (0) Line 207 and 221-222. The upper bound of Devroy et al. holds for the *excess loss*, which is the difference between the loss of the learning algorithm and the smallest loss of the functions from the class. Meanwhile, the authors are using it to upper bound the loss of the learner itself. This is true only when we are in the separable realizable situation, meaning the hypotheses class contains the BEST classifier, which achieves zero error overall. This was not explicitly stated in the setting (at least, I did not notice it). (1) Words "fiber sub-bundle" in the abstract look mysterious... (2) Lines 42-43: Devroy et al. does not contain an overview of Rademacher complexities, only VC. (3) Line 64: not clear what do the authors call "a full joint model" (4) Section 2. I had a hard time guessing the meaning of $f_p$. In the end (from the proof of Theorem 3) I conclude that in all cases $f_p(x) = P(Y=1|X=x)$. This should be stated explicitly! (5) Line 110. p(y|x) = E_p(y|x) should be replaced with p(y = 1|x) = E_p(y|x). (6) It is better to present or sketch the original definition of a fat-shattering dimension somewhere around Definition 1. (7) Line 173. How can $P$ be an element of $P_t x C$ ? (8) Equations (3) and (4): parenthesis missing, in (3) dot should be switched to comma. (9) line 212: "stronger notion of a stronger notion of..." (10) I did not get a purpose of a footnote (3) (11) lines 229-230. Better to state explicitly that this is due to probabilistic method, which is a standard tool used in these situations. Refer to the same book of Devroye, Chapter 14 if necessary. (12) Line 233. p(v_p(x)|x) should be replaced with p(y = v_p(x)|x) (13) Definition of $V_l$. Note that the r.h.s. depends on $i$ and $j$ while the l.h.s. does not. Should be fixed. (14) L.h.s. of the first inequality after line 236: the authors forgot to put $1/2^{2^m} \sum_{p\in P_{2m}}$. (15) Line 246. $\bar{z}(\bar{x})$ should be replaced with $\bar{z}_p(\bar{x})$ =======UPDATE======= I have carefully read the rebuttal. unfortunately, I am going to keep my initial decisions. The main reason for this is that I still think the assumptions considered in the paper are too strong. The authors are comparing the performance of a standard supervised algorithm to the learner which knows the structure of the problem, including the $t$ function. It is not surprising to me at all that in this case the "smart" learner will easily outperform the standard SL algorithm. In my point of view, one could hope to *learn* the $t$ function during the training, but for sure it is unrealistic to assume that the algorithms *precisely knows* $t$ in advance. Thus I think the contribution of the paper is marginal.

Confidence in this Review

3-Expert (read the paper in detail, know the area, quite certain of my opinion)


Reviewer 3

Summary

The paper addresses the multi-step learning approaches that learn the final task from breaking the problem into a sequence of several intermediate steps, as in semi-supervised learning, and considering the underlying structure (t(x)) induced by the joint distribution P. This paper gives a strong theoretical explanation using a variant of VC dimension for bounding the generalization error (in term of gap between the single-step learning and multi-step learning) with application in manifold learning and group-invariant feature learning.

Qualitative Assessment

Although I am not an expert in PAC learning, I can follow the core idea behind the paper. The paper is well-written and contains different way of understanding the notion of compatibility function used in Balcan 2005. The paper assumes that the concept class is same as the hypothesis class, can we give a similar argument (in term of the gap and \gamma-uniformly \alpha-shattering) for agnostic cases?

Confidence in this Review

1-Less confident (might not have understood significant parts)


Reviewer 4

Summary

This paper attempts to present a multi-step learning framework and provide some theoretical analyses for it.

Qualitative Assessment

This paper is not well-written. References should not appear in the abstract part. It is hard to follow this paper and difficult to understand clearly what the multi-step learning is (new manifold learning?), how it works (no pseudo-code), and which famous algorithms can be summarized into this framework. The proposed framework is not well-defined, it is hard to catch the main contribution of this paper. The experimental part is too weak and cannot provide good support to this paper. It is more interesting if previous algorithms, e.g., manifold learning, has better performance in the new framework.

Confidence in this Review

2-Confident (read it all; understood it all reasonably well)


Reviewer 5

Summary

This paper provides a theoretical framework to analyze the generalization error of multi-step algorithms, such as SSL.

Qualitative Assessment

This paper proposed a new notion of \gamma-uniform shattering for statistical models especially in multi-step learning. This new framework considers the advantages of breaking a single learning problem into a sequence of two learning algorithms. It is valuable in the sense that it provides a tentative analysis for SSL which previous theoretical analysis fails to perform. However, the paper is still limited in the following aspects. First, the paper's organization needs some further modification. I feel difficult to follow the authors explanation of both background introduction and the main results. Also, the proof for main result is too long to put in main text. It may make the paper more succinct if it is briefly explained in main text and put details in supplementary. Second, the paper lacks detailed explanation in the experimental parts, which suppose to be a very important piece in this paper. The authors just throw out lots of results without clear clarification.

Confidence in this Review

1-Less confident (might not have understood significant parts)


Reviewer 6

Summary

This paper focuses on the theoretical analysis of multi-step learning by defining the learning problem generatively as a full statistical model, which results in a natural compatibility function that agrees with the work by Balcan-Blum. In particular, they start by defining the \gamma-uniform \alpha-shattering dimension of a class of probability distributions on Xx{0,1}, and focus on the cases where the marginals p_X is uniform over S. The main contribution is in Theorem 3, which provides both the upper bound for the worst case expected loss of the best learning algorithm with oracle knowledge of the sufficient statistic, and the lower bound for the worse case expected loss of the best learning algorithm with no such knowledge. Finally, the authors apply the notion of uniform shattering in both manifold learning and invariant feature learning situations, and draw the conclusion that learning becomes impossible unless the learner has access to very large amounts of labeled data, or the learner uses a two-step semi-supervised approach.

Qualitative Assessment

The paper is well written in general, with clear contribution regarding the theoretical benefit of leveraging the unlabeled data via the sufficient statistics t_{pX}. A few questions: 1. Theorem 3 essentially applies to the cases where the VC dimension is finite. How would the result look like if this is not the case? 2. Page 2, lines 52-54, the authors stated that ‘One immediate observation is that without some tie between the possible marginals D on X and the concept class C which records possible conditionals p(y|x), there is no benefit to unlabeled data’. How is this statement supported in Theorem 3? 3. Q(m) and R(m) are derived for the cases with and without oracle knowledge of t_{pX}. If such knowledge is derived from unlabeled data, what would the worst case expected loss be?

Confidence in this Review

2-Confident (read it all; understood it all reasonably well)